# Cardiovascular Disease and Diabetes Are Among the Main Underlying Causes of Death in Twenty Healthcare Facilities Across Two Cities in the Democratic Republic of Congo

**DOI:** 10.3390/ijerph21111450

**Published:** 2024-10-31

**Authors:** Karl B. Angendu, Pierre Z. Akilimali, Dieudonné K. Mwamba, Allan Komakech, Julien Magne

**Affiliations:** 1Inserm U1094, IRD UMR270, CHU Limoges, EpiMaCT—Epidemiology of Chronic Diseases in Tropical Zone, Institute of Epidemiology and Tropical Neurology, OmegaHealth, University of Limoges, 87000 Limoges, France; karl.angendu_baki@unilim.fr (K.B.A.); julien.magne@unilim.fr (J.M.); 2The Democratic Republic of Congo National Public Health Institute, Kinshasa P.O. Box 3243, Congo; dieudonnemwambakazadi@gmail.com; 3Faculty of Medicine, Christian University of Kinshasa, Kinshasa P.O. Box 834, Congo; 4Department of Nutrition, Kinshasa School of Public Health, University of Kinshasa, Kinshasa P.O. Box 11850, Congo; 5Patrick Kayembe Research Center, Kinshasa School of Public Health, University of Kinshasa, Kinshasa P.O. Box 11850, Congo; 6Africa Centers for Disease Control and Prevention, Kinshasa P.O. Box 3243, Congo; akomackech@musph.ac.ug

**Keywords:** ICD-11, proportional mortality, underlying cause of death, diabetes, cardiovascular disease

## Abstract

Introduction: The mortality rates associated with cardiovascular disease (CVD) and diabetes exhibit disparities by region, with Central Africa ranking fourth globally in terms of mortality rate. The Democratic Republic of Congo (DRC) does not possess mortality data pertaining to these specific underlying causes of death. This study aimed to determine the death rate attributable to CVD and diabetes in two cities in the DRC. Methodology: The data on CVD and diabetes utilized in this study were obtained from a pilot project and were registered in the National Health Information System (NHIS). Data quality was initially evaluated using an automated Digital Open Rule Integrated Selection (DORIS), followed by an assessment conducted manually by three assessors. Descriptive and comparative analyses were carried out to determine the proportion of mortality related to CVD and diabetes. Results: CVD accounted for 20.4% (95%CI: 17.7–23.4%) of deaths in the two cities (Kinshasa and Matadi), whereas diabetes accounted for 5.4% (95%CI: 3.9–7.2%). After adjusting for age and city, the proportional mortality from CVD and diabetes was higher for women than men and increased with age. This study recorded 4.4% of deaths among men and 7.0% among women as the proportional mortality from diabetes. Conclusions: Non-communicable diseases (NCDs) continue to be a major cause of death, and CVD and diabetes are among the leading causes of early mortality in adults in urban areas. The proportional mortality related to CVD and diabetes appears to be higher in women than in men. Special emphasis should be placed on women, particularly during adulthood, to ensure the prompt detection of diabetes and cardiovascular conditions.

## 1. Introduction

For more than two decades, communicable diseases (CDs) (malaria, HIV, hepatitis A, B, and C, measles, salmonella, measles, blood-borne illnesses, etc.) have been recognized as the leading cause of death in sub-Saharan Africa (SSA) [1]. Non-communicable diseases (NCDs) result in 41 million fatalities annually, constituting 74% of global mortality. Seventy-seven percent of fatalities from non-communicable diseases occur in low- and middle-income countries. Cardiovascular diseases (CVDs) are responsible for most fatalities from non-communicable diseases (NCDs), totaling 17.9 million annually. Diabetes accounts for two million of these fatalities [2]. Moreover, this region is facing an emergence of CVD and diabetes, which may lead to significant burdens of morbidity and mortality. Worldwide, mortality from CVD and diabetes varies from region to region [3,4]. In 2022, the CVD mortality rate in the countries of central sub-Saharan Africa ranged from 323.5 to 464.6 per 100,000 inhabitants. Similarly, its worldwide rank increased from 9th in 1990 to 4th in 2022 [3]. In low- and middle-income countries, diabetes-related mortality has risen by 13% in two decades [5]. From 1990 to 2017, this region experienced a 67% rise in the total disability-adjusted life years (DALYs) attributable to NCDs. This augmentation is also observable in relation to other disorders. In 1990, non-communicable diseases constituted 18.6% of the overall disease burden in SSA. However, by 2017, this figure had escalated to 29.8%. Fifteen percent of Disability-Adjusted Life Years (DALYs) resulting from NCDs are ascribed to CVD [6].

To the best of our knowledge, epidemiological research in the Democratic Republic of Congo (DRC) has primarily focused on CDs. Consequently, there are currently no published studies examining the exact impact of mortality caused by CVD and diabetes. In addition, there is a scarcity of data obtained from hospital research, with only one series available that describes mortality due to hypertension (HTN) [7]. The literature extensively discusses the significant impact of globalization and urbanization on sub-Saharan Africa, notwithstanding the ongoing epidemiological change [6]. This study aimed to determine the proportion of CVD- and diabetes-related deaths in 20 healthcare facilities (HFs) located in two cities (Kinshasa and Matadi) of the DRC. The purpose of this research was to provide accurate information to policymakers and public health professionals involved in the prevention and management of these diseases. This was a pilot study and its findings do not represent the complete picture of these two cities (Kinshasa and Matadi), nor of the DRC, and even less so of Central Africa. However, Matadi and Kinshasa share borders with Congo Brazzaville and Angola.

## 2. Methods

### 2.1. Setting

This study was implemented in two cities in the DRC: Matadi, situated in the Kongo Central province with an estimated population of 448,004, and Kinshasa, with an estimated population of 17,032,322 [8]. These cities were selected for their robust internet connectivity, availability of adequate resources for us to effectively conduct mortality cause surveillance, and high notification rates of HTN and diabetes cases.

### 2.2. Data Collection and Assignment of Underlying Cause of Death

Two experts from the African Regional Office of the World Health Organization (WHO) trained twenty-two national experts in medical certification of causes of death using the 11th version of the International Classification of Diseases (ICD-11) electronic tool and verbal autopsy methods [9]. Subsequently, four national experts, with the assistance of WHO specialists, provided training to fifty individuals in the two pilot cities, including health facility personnel and experts from the two provincial health divisions.

Following the training phases, personnel at 20 focal points (FPs) from 20 health facilities (HFs) carried out the collection and transmission of data using medical death certificates configured in DHIS2’s tracker. The duration of this process was nine months, commencing in September 2022 and concluding in June 2023. The underlying cause of death refers to the specific disease, injury, accident, or violence that immediately triggered the sequence of events resulting in death [10]. Information was gathered from the medical records of 838 individuals who were admitted to and passed away in the 20 healthcare facilities (HFs) involved in the pilot project. The physicians certified these records during medical staff meetings, and the data cleaning process was conducted in three steps. Initially, the Digital Open Rule Integrated Selection (DORIS) application, developed by the WHO, was utilized to automatically identify the primary causes of death from a range of potential causes [11]. This application analyzes the data on death certificates and facilitates the automatic selection of the underlying cause of death in accordance with the fully digitalized mortality guidelines of the ICD. The methodology employed in the training sessions relied mostly on theoretical presentations, supplemented by practical activities on the majority of days. Authentic medical records obtained from a general referral hospital and a health center were utilized for these practical activities. The training themes encompassed NCDs, CDs, surgical conditions and events, as well as gynecological, obstetrical, and pediatric diseases and events. Additional factors, including accidents on public thoroughfares, poisoning, and homicide, were also considered. Practical exercises encompassed identifying causes of death, classifying them by type (underlying, intermediate, and immediate), and utilizing the ICD-11 electronic tool for their identification and selection.

Subsequently, a thorough examination of the coherence and physiological reasoning behind the sequence of causes of death was conducted by two physicians involved in project implementation as well as one who was not. Finally, any instances of repetition were eliminated. If there was a difference of opinion between the two physicians, a third, more seasoned physician was sought for consultation. Following the completion of the data cleaning procedure, a total of 778 deceased individuals were considered for this study, resulting in a final inclusion percentage of 92.8%.

To maintain data integrity during the pilot project, three national and provincial specialists coordinated two supervision missions for the FPs. These missions guaranteed that data were provided by skilled FPs and that the certification of cause of death was carried out by all medical personnel, rather than a single doctor. The accurate comprehension of the various factors leading to mortality, encompassing direct, intermediate, and underlying causes, was also confirmed by these missions [10].

### 2.3. Categorization of Deceased Persons

Prior to commencing the analysis, the deceased individuals included in the database were classified into four primary groups (Appendix A): (i) individuals with non-communicable diseases (NCDs) as an underlying cause of death, (ii) individuals with CDs or infectious diseases as an underlying cause of death, (iii) individuals with other events or diseases as an underlying cause of death, and (iv) individuals with an undetermined underlying cause of death. The group with NCDs as an underlying cause of death was further separated into four subgroups: (i) individuals with CVD as an underlying cause of death, (ii) individuals with diabetes as an underlying cause of death, (iii) individuals with kidney disease (KD) as an underlying cause of death, and (iv) individuals with other NCDs as an underlying cause of death. If the cause of death belonged to the category of disorders classified by the WHO as NCDs, it was designated as an underlying cause of death belonging to this group [2].

A CVD is defined as a condition that impacts the heart or blood arteries. This includes conditions such as coronary heart disease, cerebrovascular illnesses, peripheral arterial diseases, rheumatic heart diseases, congenital heart defects, deep vein thrombosis, pulmonary embolism, and others [12]. For the NCD group, diabetes was identified as an underlying cause when verified by diagnosis. KD encompassed both the acute and chronic forms of the disease. Other underlying causes resulting in classification within the NCD group refers to those that did not fall into any of the previously described NCD categories. Furthermore, an underlying cause of death was categorized in the CD group when the diagnosis was linked to a pathogenic agent such as bacteria, viruses, parasites, or fungus [12]. It was deemed indeterminate if there was no clear connection to the immediate cause, lack of a pathophysiological relationship, or not a condition or maternal cause of newborn death. The last category of underlying causes was for those causes that did not fall into the categories of NCD, CD, or undetermined causes.

### 2.4. Statistical Analysis

The underlying causes of death were aggregated into the following disease categories: CD, CVD, diabetes, KD, other NCD, other causes, and undetermined cause. Cause-specific proportional mortality was calculated by city, sex, and age group (<35 years, 35–59 years, and 60 years and over). Exact binomial (Clopper–Pearson) confidence intervals (95%CIs) for proportional mortality were calculated. Qualitative variables were expressed as the number and proportion (%), whereas quantitative data were presented as the median and interquartile range. The normality of the distribution of quantitative variables was assessed using the Shapiro–Wilk test. Proportional comparisons based on qualitative variables were conducted using either Pearson’s χ^2^ test or Fisher’s exact test. Multinomial logistic regression, which is used when the outcome variable being predicted is nominal and has more than two categories that do not have a given rank or order, was conducted to assess the relationship between covariates (sex, age, and city) and the underlying cause of death.

The underlying cause of death was the dependent variable, with seven modalities (communicable diseases, cardiovascular disease, diabetes, kidney disease, other non-communicable disease, other causes, and undetermined cause). We included three covariates in the model: sex, city, and age (treated as a continuous variable). All tests were conducted with a significance level of α = 0.05. All statistical analyses were performed using Stata 17 (StataCorp, College Station, TX, USA). The DORIS application was utilized to automatically identify the underlying causes of death.

### 2.5. Ethical Considerations

Approval was acquired from the health authorities of the DRC National Public Health Institute (NPHI). To ensure confidentiality, we identified the variables guaranteeing anonymity in the database. Regarding informed consent, we had no contact with the patients, and no biological procedures were involved in the collection or processing of the data; instead, the patients’ files were used posthumously to obtain the necessary information. The use of these study results will be strictly limited to purposes related to the study objectives, and the authors report no conflicts of interest. This study received ethical approval from the Ethics Committee of Kinshasa School of Public Health (KSPH) (reference number: ESP/CE/77B/2022).

## 3. Results

### 3.1. Sample Description

Following the evaluation of data quality, a total of 778 deceased individuals (median age: 45 years (13–64)) were included in this study, with 412 (54.4%) being male. Information on the deceased’s sex was lacking in three percent of cases. One in four deaths occurred in individuals under 13 years old (first quartile), 50% of the deceased were under the age of 45 at the time of death (median age), and 25% of the deaths involved individuals over the age of 64 (third quartile). There was no significant difference in age between the sexes (45 years (12–63.7) in men vs. 47 years, (16–64) in women, *p* = 0.421). Approximately 14% of the recorded cases, namely 110 out of 778, did not include age information. According to the distribution analysis, the cases where age information was missing were overall similar to those for which age information was available in terms of sex as well as for the two cities (Table 1).

### 3.2. Sex and City Breakdown of Case Distribution of Underlying Causes

A total of 578 cases occurred in Kinshasa and 200 in Matadi. Three out of four cases in the sample were recorded in Kinshasa. The distribution of causes exhibited non-uniformity with respect to sex and cities. The proportional mortality from communicable diseases decreased with age (Figure 1). Matadi has a higher proportion of KD as an underlying cause of death, accounting for 4% of all deaths, compared to only 1% in Kinshasa (Figure 2). CVD was identified as a noteworthy contributor to mortality in women relative to men (Table 2 and Table 3).

### 3.3. Cardiovascular Disease Mortality

Based on underlying cause of death certification, CVD caused 20.4% (95%CI: 17.7–23.4%) of deaths in the two cities. After adjustment for age and city, the proportional mortality from CVD was higher in women than in men. For this study, 18.9% (95%CI: 15.3–23.1%) of deaths among men and 23.5% (95%CI: 19.1–28.3%) of deaths among women (Table 4) were recorded as being from CVD. The proportional mortality from CVD increased with age (Figure 3). In older adults, aged 60 years and over, the proportional mortality from CVD was 41.8% (95%CI: 35.1–48.7%) higher than that in adults aged from 35 to 59 (28.6%). The death rates for cardiovascular disease and diabetes in the two cities were indistinguishable, as their confidence intervals overlapped.

### 3.4. Diabetes Mortality

This study found that women who died from diabetes had a higher average age at death compared to men (63.1 vs. 59.3; *p* = 0.255). However, this difference was not statistically significant. Based on the underlying cause of death certification, diabetes accounted for 5.4% (95%CI: 3.9–7.2%) of deaths in the two cities. The proportional mortality from diabetes increased with age. This study recorded 4.4% (95%CI: 2.6–6.8%) of deaths among men and 7.0% (95%CI: 4.5–10.2%) of deaths among women as being from diabetes (Table 4). After adjustment for age and city, the proportional mortality from diabetes was higher among women compared to men (Figure 4).

## 4. Discussion

To the best of our knowledge, this study is the first to estimate cause-specific proportional mortality related to CVD and diabetes in the DRC. Complementing another study carried out by our team, the present study gives an idea of the mortality burden represented by these diseases. While communicable diseases continue to be the main underlying cause of death, non-communicable diseases such as diabetes and CVD are also major contributors to deaths in both urban areas. Twenty percent of deaths are due to CVD and 5% are due to diabetes. The risk of death from these two diseases increases with age. In this study, a difference based on sex was observed in terms of proportional mortality related to CVD and diabetes, which appeared higher in women than in men. Limited research on mortality from NCDs in SSA has employed a methodology akin to that utilized in our study. In one exception, a Tanzanian study approximated hospital mortality for diabetes at 3.8% and for cardiovascular diseases at nearly 25% [13]. Like our study, the researchers found that for these diseases, older and female subjects accounted for an oversized proportion of mortality.

The incidence and prevalence of type 2 diabetes have risen globally, with elevated rates observed in low–middle, middle, and high–middle Socio-demographic Index nations. The rising incidence of type 1 diabetes has primarily transpired in high-income areas, notably Europe and the United States, where a yearly increase of 2.7–4.0% in type 1 diabetes cases has been documented [14,15,16]. This suggests that individuals in low–middle, middle, and high–middle Socio-demographic Index countries may be more susceptible to type 2 diabetes due to social and economic changes, characterized by an increasing food supply, a westernized diet, and less physical activity. Regional disparities can be partially ascribed to elevated rates of chronic illness, poverty, disjointed healthcare, and insufficient access to both preventive and specialized care in rural regions [17].

The results presented in this study have significance for improving strategies to prevent and manage the leading causes of early adult death in the cities of Kinshasa and Matadi—the capital of Kongo Central province. The 2023 Demographic and Health Survey [18] revealed that the prevalence of hypertension was 11% among women and 13% among men in Kinshasa, and it was 11% among women and 22% among males in Kongo Central. The survey also found a prevalence of diabetes of 10% among women and 7% among men in Kinshasa, and of 2% among women and 5% among men in Kongo Central.

There are a greater number and variety of modifiable risk factors for atherosclerotic cardiovascular disease (including stroke) than for adult-onset diabetes mellitus (type 2 diabetes). These risk factors include HTN, tobacco smoking, and dyslipidemia [19]. There is compelling evidence supporting the effectiveness of interventions that focus on these risk factors, both in individuals [20,21,22] and in populations [23,24], with the purpose of preventing and controlling diseases. The prevention and control of CVD places significant emphasis on dietary measures such as reducing saturated and trans fats as well as reducing salt intake. Additionally, efforts are being expended to address tobacco use. This is because the reversal of diabetes and maintaining optimal control of blood sugar levels is challenging, and the evidence regarding its effectiveness is inconclusive [25,26]. CVD is a leading underlying cause of mortality among individuals with diabetes.

A significant potential outcome of artificially inflated diabetes mortality rates is an excessive emphasis on addressing diabetes prevention and control, rather than a more comprehensive approach to addressing CVD risks. In contexts where the health system lacks resources, the observed decrease in CVD mortality, whose causes are not yet understood, may lead to reductions in expenditure and initiatives aimed at preventing CVD.

Sex-based differences in cause-specific proportional mortality related to CVD and diabetes were observed in the present study. Increasing evidence suggests that sex disparities play a significant role in the etiology and treatment of various diseases, with certain characteristics having a notable impact on the outcomes of severe illnesses. Sex differences refer to the biological variations between females and males. These differences are influenced by variations in the genetic information carried by sex chromosomes, the specific gene expression of autosomes that are linked to sex, the varying levels and effects of sex hormones, and the distinct target organs that these hormones act upon. Moreover, both sexes have metabolic alterations throughout their lifespans, with women exhibiting more pronounced changes as a result of their reproductive function. Diabetologists have long recognized that diabetes significantly impacts cardiovascular health in women, regardless of the type of diabetes [27]. CVD is the primary cause of morbidity and mortality in patients with diabetes, responsible for almost 50% of all deaths [28]. Although sex-specific differences are garnering a heightened focus in cardiology, the underlying processes causing this link to remain ambiguous. The pathogenesis appears to be multifaceted, influenced by genetic and biological sex differences, cultural and environmental sex disparities, and the established variances in the diagnosis, management, and treatment of diabetes mellitus and cardiovascular disease between women and men [29,30]. Women with diabetes are more prone to unfavorable risk factor profiles and experience a heightened disease risk due to the impact of specific risk factors. A recent meta-analysis indicated that smoking presented a 25% greater risk for coronary heart disease in women compared to men [31]. Furthermore, women with diabetes are less likely to meet high-density lipoprotein cholesterol objectives and have a greater prevalence of obesity compared to men [32]. Sex differences arise from a combination of genetic factors and socio-cultural factors, such as habits, behaviors, lifestyles, exposure to environmental influences, dietary and lifestyle choices, stress levels, and attitudes toward treatments and disease prevention campaigns. Sex disparities can influence behavior across the lifespan, and physiological transformations can have consequences for one’s way of life, societal responsibilities, and psychological well-being. A previous study, which examined 26 diabetes studies from as early as 1965, also discovered that women had a higher risk of dying from heart disease than men, which, in this case, was more than twice as high [33]. The relative risk of suffering from heart failure in women with diabetes compared to those without diabetes is higher than in men [34]. The authors hypothesize that the disparity in glycemic control between girls and women compared to boys and men may be attributed to continuous poor management of blood sugar levels. This could be attributable to a more pronounced decline in insulin sensitivity during puberty in young women with type 1 diabetes. In the context of the DRC, the disparity in mortality rates from cardiovascular disease and diabetes between sexes could be explained by socio-cultural, educational, and economic factors and lifestyle choices. In the DRC, women typically possess less education compared to men. They mostly utilize anabolic substances in an effort to augment their physical mass. Based on these characteristics identified as probable drivers of elevated mortality in women, the following public health measures should be contemplated: increasing awareness of the necessity of regular physical activity, nutritious dietary practices, and the adoption of health-enhancing behaviors. Mass screening initiatives for non-communicable diseases should be contemplated as well. Recent studies indicate that women are more susceptible than men to experiencing eating disorders and underdosing on insulin. According to the American Diabetes Association, women with diabetes of any kind have a higher likelihood of experiencing eating disorders compared to women without diabetes [35]. Among women with type 1 diabetes, bulimia is the most prevalent eating disorder [35]. Additionally, compared to men, women are more prone to having undiagnosed or untreated underlying risk factors or health issues and are at higher risk of under-recognition and under-treatment of heart disease. While both sexes are at risk when certain lifestyle factors and health conditions are present, such as high cholesterol, high blood pressure, smoking, and obesity, there are specific reasons as to why heart disease in women may remain undetected and untreated and result in disease progression. This implies that illnesses such as HTN, hyperlipidemia, or diabetes proceed to more advanced stages, resulting in a worse outcome when they are eventually treated or leading to a heart attack [35]. On average, women tend to be older than men when experiencing a heart attack, with women often being around 70 years old, which is over 4 years older than for men [35]. Prior research has indicated that women diagnosed with ST-elevation myocardial infarction experience a more unfavorable prognosis while hospitalized in comparison to men [35]. This disparity may be attributed to their advanced age, higher prevalence of comorbidities, and lower utilization of stents (percutaneous coronary intervention (PCI)) for the purpose of restoring blood flow in blocked arteries [35].

Regarding certification practices, likely contributors to certifier reporting of diabetes and the increasing secular trend in diabetes mortality in the two cities include the following: increased testing and diagnosis of diabetes; government awareness of diabetes; and the development partner activities [36]. The heightened susceptibility to and prevalence of CVD and associated risk factors in individuals with diabetes [36] can complicate the process of certifying diabetes as the cause of death [37,38]. Certifiers may possess varying interpretations of the causal connections between diabetes and CVD, as well as the risk factors associated with CVD and other medical diseases.

In relation to coding norms, the misclassification of simple and non-fatal diabetes may lead to diabetes being incorrectly identified as the underlying cause of death. This is owing to inconsistencies in the ICD-11 coding regulations pertaining to diabetes and its complications. The ICD-11 coding rules acknowledge that common CVDs, such as ischemic heart disease (IHD), hypertensive diseases, and other conditions, can be attributed to diabetes if they are documented. This includes diabetes with non-fatal complications, diabetes without complications, or diabetes with unspecified complications [39,40]. These principles appear incongruous with the prevailing medical knowledge, save for the infrequent occurrence of diabetic cardiomyopathy [41].

The limitations of the present study include the potential for overestimating or underestimating diabetes mortality in the alternative cause-of-death certification. Misclassification or miscoding on death certificates is possible, and not listing diabetes as a contributing cause may have led to us underestimating the burden of diabetes-related mortality. However, the use of the DORIS application and a manual check resulted in a database with consistent underlying causes. The lack of standard guides on disease definitions for all the focal points could have led to information bias. This bias was mitigated by using teams of medical staff for diagnoses during the medical certification of causes of death, rather than relying on a single doctor. A selection bias could also have been present in this study, as only data from the HFs and not from the community were taken into account. This pilot study, encompassing two cities and 20 HFs, does not necessarily reflect the entirety of HFs in the DRC, nor does it account for deaths occurring inside communities. The causes mentioned in this study do not necessarily reflect the actual causes of mortality in the DRC. Using hospital-based data versus community data particularly in the context of underreporting in rural or less-connected areas can be also listed as one of the limitations. Another limitation of this study is that the sequence of disease onset facilitated the identification of the early pathology as the primary cause of mortality. A comprehensive national study is essential to accurately depict the causes of mortality across the country.

For some cases in this study, the precise attribution of the underlying causes of death was lacking, and these were categorized as indeterminate cases. The underlying cause of death was not classified in about 13% of cases, which likely led to a slight underestimation of mortality related to diabetes and CVD. The possible substitution of diabetes for CVD as an underlying cause of death due to certification and coding artifacts has been demonstrated elsewhere [24,42]. The former includes cases of unspecified chronic renal disease or chronic renal failure coincident with diabetes that may not have been caused by diabetes. Thus, the inclusion of diabetes may result in the overestimation of ‘diabetes with renal complications’ as the underlying cause of death. Moreover, other plausible causes of death may not have been recorded, leaving diabetes as the only recorded condition. An example of the possible underestimation of diabetes mortality is that the certifier may have recorded ‘diabetes’ without also recording the occurrence of potentially fatal diabetic complications. More complete recording and better characterization of disease-specific complications are needed. Finally, our analysis focused on only two cities, which are not representative of the whole country.

This study used hospital discharge data to strengthen the validity and accuracy of the final underlying causes of death assignment. It is the first to have reviewed the causes of death in the DRC and use automated ICD-11 coding to efficiently standardize the assignment of the underlying causes of death. The distribution analysis indicated that instances lacking age information were generally comparable to those with available age data for both sexes and the two cities. Nonetheless, the distribution of underlying causes of mortality varied, and it is understood that the cause of death correlates with age. This may have resulted in selection bias and skewed the trend of underlying causes of death by age.

The results of this study can serve as a basis for guiding the development of standards, guidelines, procedures, and plans aimed at ultimately reducing the burden of NCD-related mortality. Public health actors who engage with large communities should be made aware of the high mortality linked to NCDs so that they can integrate NCD-related themes into their mass activities.

## 5. Conclusions

NCDs continue to be a major cause of death, and CVD and diabetes are among the leading causes of early mortality in adults in urban areas. The proportional mortality related to CVD and diabetes appears to be higher among women than men. Special emphasis should be placed on women, particularly during adulthood, to ensure the prompt detection of diabetes and cardiovascular conditions. The omission of potentially life-threatening complications when reporting diabetes may have resulted in an overestimation of diabetes-related mortality and an underestimation of mortality due to CVD in the two examined cities. To ensure the accurate assignment of underlying causes of mortality for public health purposes, it is necessary to have a precise certification of diabetes as either a direct cause or a contributory cause. Further studies should also look at the national level and use representative sampling, to estimate the cause-specific proportional mortality related to CVD and diabetes and provide an accurate picture of causes of death in the DRC.

## Figures and Tables

**Figure 1 ijerph-21-01450-f001:**
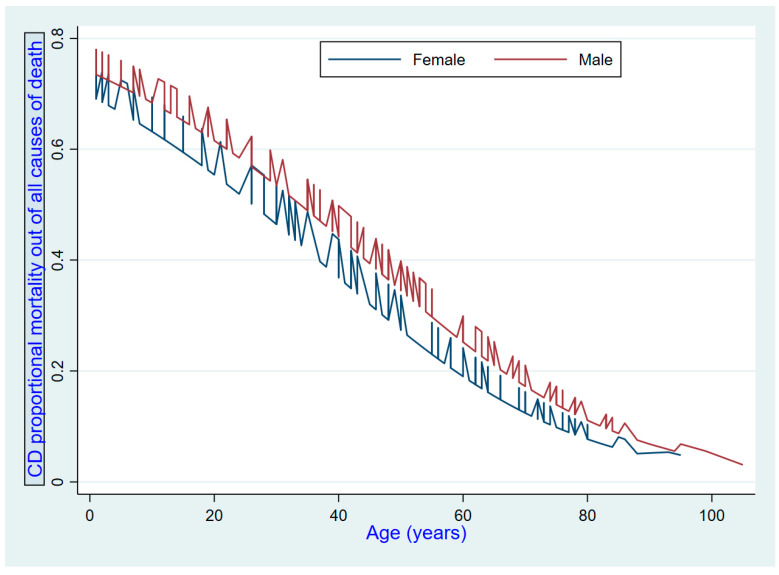
Trend of proportional mortality from communicable diseases (CD) by age and sex.

**Figure 2 ijerph-21-01450-f002:**
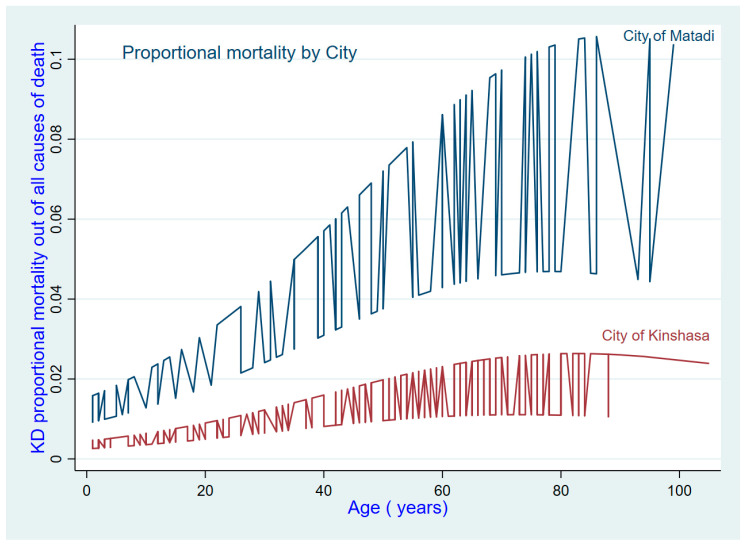
Trend of proportional mortality from kidney disease (KD) by age and city.

**Figure 3 ijerph-21-01450-f003:**
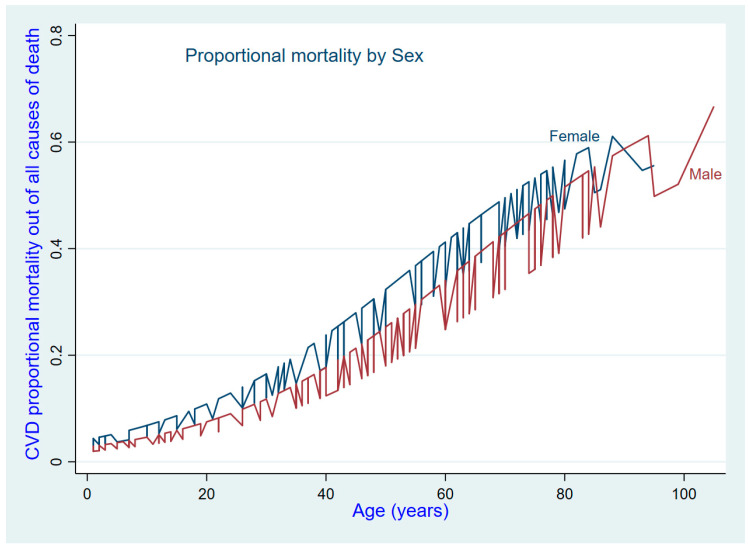
Trend of proportional mortality from cardiovascular disease by age and sex.

**Figure 4 ijerph-21-01450-f004:**
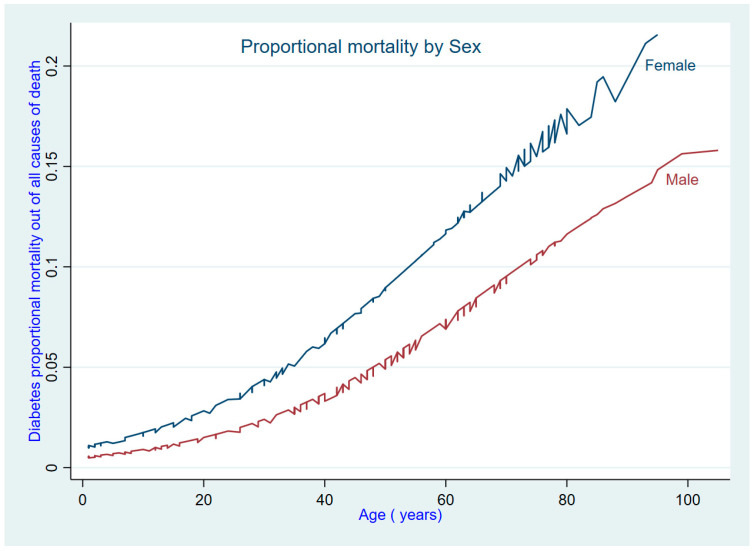
Trend of proportional mortality from diabetes by age and sex.

**Table 1 ijerph-21-01450-t001:** Characteristics of samples with missing age information data.

	Data on Age						*p*
	Available		Missing		Total		
	*n*	%	*n*	%	*n*	%	
Sex							0.162
Female	291	44.6	54	51.9	345	45.6	
Male	362	55.4	50	48.1	412	54.4	
Total	653	100.0	104	100.0	757	100.0	
Underlying causes of death							<0.001
Communicable diseases	249	37.3	48	43.6	297	38.2	
Cardiovascular disease	158	23.7	1	0.9	159	20.4	
Diabetes	42	6.3	0	0.0	42	5.4	
Kidney diseases	15	2.2	0	0.0	15	1.9	
Other non-communicable disease	51	7.6	1	0.9	52	6.7	
Other causes	101	15.1	15	13.6	116	14.9	
Undetermined causes	52	7.8	45	40.9	97	12.5	
Total	668	100.0	110	100.0	778	100.0	
City							0.266
Matadi	167	25.0	33	30.0	200	25.7	
Kinshasa	501	75.0	77	70.0	578	74.3	
Total	668	100.0	110	100.0	778	100.0	

**Table 2 ijerph-21-01450-t002:** Distribution of sample by city.

	Matadi (*n* = 200)		Kinshasa (*n* = 578)		Total (*n* = 778)		*p*
	*n*	%	*n*	%	*n*	%	
Age, median [Q1–Q3]	45 [23–65]		45 [10–64]		45 [13–64]		
Age (years)							0.253
<5	19	11.4	97	19.4	116	17.4	
5–11	9	5.4	30	6.0	39	5.8	
12–19	10	6.0	31	6.2	41	6.1	
20–34	18	10.8	52	10.4	70	10.5	
35–59	55	32.9	134	26.7	189	28.3	
≥60	56	33.5	157	31.3	213	31.9	
Total	167	100.0	501	100.0	668	100.0	
Sex							0.873
Female	88	46.1	257	45.4	345	45.6	
Male	103	53.9	309	54.6	412	54.4	
Total	191	100.0	566	100.0	757	100.0	
Underlying causes of death							0.009
Communicable diseases	70	35.0	227	39.3	297	38.2	
Cardiovascular disease	34	17.0	125	21.6	159	20.4	
Diabetes	11	5.5	31	5.4	42	5.4	
Kidney diseases	9	4.5	6	1.0	15	1.9	
Other non-communicable disease	18	9.0	34	5.9	52	6.7	
Other causes	26	13.0	90	15.6	116	14.9	
Undetermined	32	16.0	65	11.2	97	12.5	
Total	200	100.0	578	100.0	778	100.0	

**Table 3 ijerph-21-01450-t003:** Distribution of sample by sex.

	Female (*n* = 345)		Male (*n* = 412)		Total (*n* = 757)		
	*n*	%	*n*	%	*n*	%	
Age (years)							0.496
<5	46	15.8	69	19.1	115	17.6	
5–11	16	5.5	21	5.8	37	5.7	
12–19	16	5.5	22	6.1	38	5.8	
20–34	36	12.4	30	8.3	66	10.1	
35–59	79	27.1	107	29.6	186	28.5	
≥60	98	33.7	113	31.2	211	32.3	
Total	291	100.0	362	100.0	653	100.0	
Underlying causes of death							0.048
Communicable diseases	123	35.7	169	41.0	292	38.6	
Cardiovascular disease	81	23.5	78	18.9	159	21.0	
Diabetes	24	7.0	18	4.4	42	5.5	
Kidney diseases	4	1.2	10	2.4	14	1.8	
Other non-communicable disease	16	4.6	34	8.3	50	6.6	
Other causes	52	15.1	63	15.3	115	15.2	
Undetermined	45	13.0	40	9.7	85	11.2	
Total	345	100.0	412	100.0	757	100.0	
City							0.873
Matadi	88	25.5	103	25.0	191	25.2	
Kinshasa	257	74.5	309	75.0	566	74.8	
Total	345	100.0	412	100.0	757	100.0	

**Table 4 ijerph-21-01450-t004:** Proportional mortality (%) from cardiovascular disease and diabetes.

	Cardiovascular Disease		Diabetes	
	Proportional Mortality, %	95%CI *	Proportional Mortality, %	95%CI *
Age (years)				
<35	5.6	3.2–9.1	1.5	0.4–3.8
≥35	35.6	30.9–40.5	9.4	6.8–12.7
35 to 59	28.6	22.2–35.6	6.3	3.3–10.8
≥60	41.8	35.1–48.7	12.2	8.1–17.4
Sex				
Female	23.5	19.1–28.3	7.0	4.5–10.2
Male	18.9	15.3–23.1	4.4	2.6–6.8
City				
Matadi	17.0	12.1–22.9	5.5	2.8–9.6
Kinshasa	21.6	18.3–25.2	5.4	3.7–7.5
Total	20.4	17.7–23.4	5.4	3.9–7.2

*: with exact binomial (Clopper–Pearson) confidence intervals.

## Data Availability

The mortality dataset analyzed in the current study is not publicly available due to ethical restrictions (contains personal identifiable information) and is stored on secure servers with restricted access. Access was obtained through the approval of the research ethics committee and data custodians at the Ministry of Health, DRC. Data are, however, available from the corresponding author on reasonable request and with permission of the Ministry of Health, DRC.

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
