# Peer review of "Cardiovascular Disease and Diabetes Are Among the Main Underlying Causes of Death in Twenty Healthcare Facilities Across Two Cities in the Democratic Republic of Congo"

_ijerph, 2024, doi:10.3390/ijerph21111450_

Round 1
Reviewer 1 Report
Comments and Suggestions for Authors
The X axis legend in Figure 1 is not very obvious.
The inclusion and exclusion criteria for this study are not discussed by the author in this article.
Is there a relationship between the two cities' mortality rates for cardiovascular disease, non-communicable disease, and Diabetes?
Is there any information regarding the individual who experienced more than one disease?
Author Response
Comments and Suggestions for Authors
The X axis legend in Figure 1 is not very obvious.
Authors response: We have improved the quality of the figure based on this comment.
The inclusion and exclusion criteria for this study are not discussed by the author in this article.
Authors response: We thank the reviewer for this comment, and this was addressed in the current version. Which can be now read as follows in discussion section among limitation:
“This pilot study, encompassing two cities and 20 hospitals, does not necessarily reflect the entirety of healthcare institutions in the DRC, nor does it account for deaths occurring inside communities. The causes mentioned in this study do not necessarily reflect the actual causes of mortality in the DRC. “
Is there a relationship between the two cities' mortality rates for cardiovascular disease, non-communicable disease, and Diabetes?
Authors response: The death rates for cardiovascular disease and diabetes in the two cities were indistinguishable, as their confidence intervals overlapped.
Is there any information regarding the individual who experienced more than one disease?
Authors response: Another limitation of this study is that the sequence of disease onset facilitated the identification of the early pathology as the primary cause of mortality
Reviewer 2 Report
Comments and Suggestions for Authors
The necessary corrections have been marked on the article. It is appropriate to publish after the corrections are made.

Author Response
Comments and Suggestions for Authors
The necessary corrections have been marked on the article. It is appropriate to publish after the corrections are made.
Authors response: Thank you for your comment. We made requested edits.
Reviewer 3 Report
Comments and Suggestions for Authors The authors conducted a survey and analysis focusing on the proportion of deaths in the Democratic Republic of the Congo due to CVD and diabetes. The results themselves will be useful information for reducing premature mortality in adults and for formulating health management strategies in the DRC. 1. In the introduction, it was stated that there are significant regional differences in mortality rates due to CVD and diabetes. I think that this is certainly true. However, that is precisely why it is necessary to cite mortality rates due to CVD and diabetes in other countries and regions in the discussion section and discuss the differences. The same applies to differences based on gender. In this way, it should be possible to develop a deeper discussion, such as why regional differences can occur. 2. The cases where the cause of death could not be identified and the cases where the age was unknown were analysed and discussed separately, so this is appropriate. However, the description of the limitations and constraints of the research in the discussion section is verbose, so it would be better to write it more compactly.Specifically, it would be better to summarise the following parts more compactly. P.10 'In relation to coding norms, ..... Study limitations include the potential for overestimating or underestimating ......CVD and diabetes mortality on a national scale [24]' 3. The statement ‘One in five deaths is due to CVD, and 5% are due to diabetes.’ should be unified to either a statement of the number of people or a statement of the percentage.
Author Response
Comments and Suggestions for Authors
The authors conducted a survey and analysis focusing on the proportion of deaths in the Democratic Republic of the Congo due to CVD and diabetes. The results themselves will be useful information for reducing premature mortality in adults and for formulating health management strategies in the DRC. 1. In the introduction, it was stated that there are significant regional differences in mortality rates due to CVD and diabetes. I think that this is certainly true. However, that is precisely why it is necessary to cite mortality rates due to CVD and diabetes in other countries and regions in the discussion section and discuss the differences. The same applies to differences based on gender. In this way, it should be possible to develop a deeper discussion, such as why regional differences can occur.
Authors response: The mortality rates in other countries and regions was documented. This can be read as follows in discussion section: “Non-communicable diseases (NCDs) result in 41 million fatalities annually, constituting 74% of global mortality. Seventy-seven percent of fatalities from non-communicable diseases occur in low- and middle-income countries. Cardiovascular diseases (CVDs) are responsible for most fatalities from non-communicable diseases (NCDs), totaling 17.9 million annually. Diabetes accounts for two million of these fatalities. The incidence and prevalence of type 2 diabetes have risen globally, with elevated rates observed in low-middle, middle, and high-middle Socio-demographic Index nations. The rising incidence of type 1 diabetes has primarily transpired in high-income areas, notably Europe and the United States, where a yearly increase of 2.7–4.0% in type 1 diabetes cases has been documented. This suggests that individuals in low-middle, middle, and high-middle Socio-demographic Index countries may be more susceptible to type 2 diabetes due to social and economic changes, characterized by an increasing food supply, a westernized diet, and less physical activity. Regional disparities can be partially ascribed to elevated rates of chronic illness, poverty, disjointed healthcare, and insufficient access to both preventive and specialized care in rural regions”
The gender and regional differences were discussed in the discussion section. This can be read as follows in the discussion section:
“Diabetologists have long recognized that diabetes significantly impacts cardiovascular health in women, regardless of the type of diabetes. CVD is the primary cause of morbidity and mortality in patients with diabetes, responsible for almost 50% of all deaths[reference]. Although sex-specific differences are garnering heightened focus in cardiology, the underlying processes causing this link to remain ambiguous. The pathogenesis appears to be multifaceted, influenced by genetic and biological sex differences, cultural and environmental gender disparities, and the established variances in the diagnosis, management, and treatment of diabetes mellitus and cardiovascular disease between women and men. Women with diabetes are more prone to unfavorable risk factor profiles and experience heightened disease risk due to the impact of specific risk factors. A recent meta-analysis indicated that smoking presented a 25% greater risk for coronary heart disease in women compared to males. Furthermore, women with diabetes are less likely to meet high-density lipoprotein cholesterol objectives and have a greater prevalence of obesity compared to men.”
- The cases where the cause of death could not be identified and the cases where the age was unknown were analysed and discussed separately, so this is appropriate. However, the description of the limitations and constraints of the research in the discussion section is verbose, so it would be better to write it more compactly. Specifically, it would be better to summarise the following parts more compactly. P.10 'In relation to coding norms, ..... Study limitations include the potential for overestimating or underestimating ......CVD and diabetes mortality on a national scale [24]'
Authors response: Thanks for this comment, we have rewritten this section.
“This pilot study, encompassing two cities and 20 HFs, does not necessarily reflect the entirety of HFs in the DRC, nor does it account for deaths occurring inside communities. The causes mentioned in this study do not necessarily reflect the actual causes of mortality in the DRC. Another limitation of this study is that the sequence of disease onset facilitated the identification of the early pathology as the primary cause of mortality. A comprehensive national study is essential to accurately depict the causes of mortality across the country.”
The statement ‘One in five deaths is due to CVD, and 5% are due to diabetes.’ should be unified to either a statement of the number of people or a statement of the percentage.
Authors response: Thanks for this comment, “Twenty percent of deaths are due to CVD, and 5% are due to diabetes.”
Reviewer 4 Report
Comments and Suggestions for Authors
1. The abstract does not mention any specific findings regarding sex-based differences in mortality, which are emphasized later in the results. Including a brief mention of this in the abstract would give a more complete picture of the study's findings.
2. The introduction contains some repetitive phrasing, particularly in the first and second paragraphs, where global and regional CVD statistics are mentioned multiple times. Consider streamlining this section to avoid redundancy and improve readability.
3. In some sections, complex medical jargon is used without explanation, such as "multinomial logistic regression." Consider providing a brief explanation for readers who may not be familiar with these terms, especially since this manuscript could have a broad public health audience.
4. Figures 1 and 2: The legends need to be more detailed. Currently, they do not provide enough information about what the data represents.
5. In the discussion, gender differences are discussed, but the terms “gender” and “sex” are used interchangeably. Be consistent in using the correct term based on whether you refer to biological differences (sex) or socially constructed roles (gender).
6. The conclusion is generally well-stated but could be strengthened by including a call for specific future actions based on the study’s findings, such as targeting specific interventions to reduce CVD and diabetes mortality in women.
7. Some of the references are outdated, such as the Global Burden of Disease Study reference from 1997. Where possible, it would be better to cite more recent studies, particularly for key statistics on CVD and diabetes mortality.
8. The methodology section lacks clarity regarding the process of data cleaning and inclusion/exclusion criteria. It would be beneficial to include more details on the DORIS tool and how manual checks were performed by the three assessors. Additionally, consider providing more specifics about the training that health personnel received for certification, as this is crucial for ensuring accurate cause-of-death reporting.
9. The study indicates that age data is missing for 14% of cases. It is important to provide an explanation of how this missing data might impact the findings, particularly in terms of proportional mortality analysis. A more thorough discussion of potential biases resulting from this missing information should be included in both the methods and discussion sections.
10. While multinomial logistic regression is mentioned, the rationale behind using this statistical method could be expanded. Additionally, the manuscript would benefit from a brief explanation of why proportional mortality was calculated for specific age groups and how these age categories were determined.
11. The results section highlights differences in mortality between men and women, but it does not provide a clear explanation of why these differences might exist, particularly in relation to the local context of the DRC. Consider expanding the discussion to include possible sociocultural, economic, or healthcare access-related reasons for the observed differences.
12. The study acknowledges limitations, such as misclassification and missing data, but these points could be more thoroughly elaborated. The authors should consider discussing the impact of using hospital-based data versus community data and how this might skew the results, particularly in the context of underreporting in rural or less-connected areas.
Comments on the Quality of English Language
Overall Comments on the Quality of English Language:
The manuscript is generally well-written but requires moderate editing to improve readability and clarity. The main areas needing attention are sentence structure, repetition, word choice, transitions between ideas, and minor grammatical issues. Enhancing these areas will significantly improve the flow of the paper and ensure a more polished and professional tone. Below are examples illustrating the areas that require improvement:
1. Many sentences are long and complex, making them harder to follow. Simplifying and shortening these sentences would improve clarity.
Example: "The mortality rates associated with cardiovascular disease and diabetes exhibit disparities by region, with Central Africa ranking fourth globally in terms of mortality rate, this study aimed to determine the death rate attributable to CVD and diabetes in two cities in the DRC."
Suggested Revision: "Cardiovascular disease and diabetes mortality rates vary significantly by region, with Central Africa ranking fourth globally. This study aimed to determine the death rate attributable to CVD and diabetes in two cities in the DRC."
2. Certain ideas are repeated unnecessarily, which affects the conciseness of the manuscript. Removing redundant information will improve the flow. Example: "The study found that CVD caused 20.4% of deaths in the two cities. After adjusting for age and city, CVD caused 20.4% of deaths in both Kinshasa and Matadi." Suggestion: "CVD accounted for 20.4% of deaths in the two cities. This proportion remained the same after adjusting for age and city."
3. Some terms and phrases could be more precise and scientific. Using more concise and appropriate terminology will improve the manuscript's professionalism. Example: "This data was then cleaned and processed by using the DORIS tool, which is a tool designed for automatic cause-of-death identification." Revision: "The data was cleaned and processed using the DORIS tool, designed for automated cause-of-death identification."
4. Flow: The transitions between ideas, particularly in the results and discussion sections, could be smoother. Improved flow will help readers better follow the narrative. Original: "The results indicated a higher mortality rate for women. Cardiovascular diseases were a significant cause of death in the sample. Diabetes was also a major contributor." Suggested Revision: "The results showed that women had a higher mortality rate than men, with cardiovascular diseases being a significant cause of death. Diabetes was also identified as a major contributor."
Author Response
Comments and Suggestions for Authors
- The abstract does not mention any specific findings regarding sex-based differences in mortality, which are emphasized later in the results. Including a brief mention of this in the abstract would give a more complete picture of the study's findings.
Authors response: We have edited it in the abstract and, this can be seen as follows:
“This study has recorded 4.4% of deaths among men and 7.0% among women, as proportional mortality from diabetes. »
- The introduction contains some repetitive phrasing, particularly in the first and second paragraphs, where global and regional CVD statistics are mentioned multiple times. Consider streamlining this section to avoid redundancy and improve readability.
Authors response: We have edited this section accordingly
- In some sections, complex medical jargon is used without explanation, such as "multinomial logistic regression." Consider providing a brief explanation for readers who may not be familiar with these terms, especially since this manuscript could have a broad public health audience.
Authors response: We have provided short explanation which can be seen as follows: “Multinomial logistic regression, which is used when the outcome variable being predicted is nominal and has more than two categories that do not have a given rank or order”,
- Figures 1 and 2: The legends need to be more detailed. Currently, they do not provide enough information about what the data represents.
Authors response: We have improved the quality of the figures 1 and 2 based on this comment
- In the discussion, gender differences are discussed, but the terms “gender” and “sex” are used interchangeably. Be consistent in using the correct term based on whether you refer to biological differences (sex) or socially constructed roles (gender).
Authors response: We have edited accordingly, and we are using only sex in the text.
- The conclusion is generally well-stated but could be strengthened by including a call for specific future actions based on the study’s findings, such as targeting specific interventions to reduce CVD and diabetes mortality in women.
Authors response: We have edited accordingly and now in the conclusion, this statement was added: “Further studies should also look at national level using a representative sampling, to estimate cause-specific proportional mortality related to CVD and diabetes and provide accurate picture of causes of death in DRC.”
- Some of the references are outdated, such as the Global Burden of Disease Study reference from 1997. Where possible, it would be better to cite more recent studies, particularly for key statistics on CVD and diabetes mortality.
Authors response: Thank you for your comment and we have cited the most recent study: “Standing up to infectious disease. Nat Microbiol. 2019 Jan;4(1):1. doi: 10.1038/s41564-018-0331-3. PMID: 30546101; PMCID: PMC7097104”.
- The methodology section lacks clarity regarding the process of data cleaning and inclusion/exclusion criteria. It would be beneficial to include more details on the DORIS tool and how manual checks were performed by the three assessors. Additionally, consider providing more specifics about the training that health personnel received for certification, as this is crucial for ensuring accurate cause-of-death reporting.
Authors response: The Digital Open Rule Integrated Selection (DORIS) analyzes the data on death certificates and facilitates the automatic selection of the underlying cause of death in accordance with the fully digitalized mortality guidelines of the ICD. The other point was addressed in the current version.
- The study indicates that age data is missing for 14% of cases. It is important to provide an explanation of how this missing data might impact the findings, particularly in terms of proportional mortality analysis. A more thorough discussion of potential biases resulting from this missing information should be included in both the methods and discussion sections.
Authors response: We have edited accordingly and added this statement: “The distribution analysis indicates that instances lacking age information were generally comparable to those with available age data for sex and between the two cities. Nonetheless, the distribution of underlying causes of mortality varied, and it is understood that the cause of death correlates with age. This may result in selection bias and skew the trend of underlying causes of death by age.”
- While multinomial logistic regression is mentioned, the rationale behind using this statistical method could be expanded. Additionally, the manuscript would benefit from a brief explanation of why proportional mortality was calculated for specific age groups and how these age categories were determined.
Authors response: We have provided short explanation which can be seen as follows: “Multinomial logistic regression, which is used when the outcome variable being predicted is nominal and has more than two categories that do not have a given rank or order”, The outcome variable Underlying causes of death has seven categories: Communicable diseases, Cardiovascular disease, Diabetes, Kidney diseases, Other non-communicable disease, Other causes and Undetermined causes. From Multinomial logistic regression, post estimation provided figures 1, 2, 3 and 4.
- The results section highlights differences in mortality between men and women, but it does not provide a clear explanation of why these differences might exist, particularly in relation to the local context of the DRC. Consider expanding the discussion to include possible sociocultural, economic, or healthcare access-related reasons for the observed differences.
Authors response: We have expanded discussion by including possible socio-cultural , economic etc… considerations.
- The study acknowledges limitations, such as misclassification and missing data, but these points could be more thoroughly elaborated. The authors should consider discussing the impact of using hospital-based data versus community data and how this might skew the results, particularly in the context of underreporting in rural or less-connected areas.
Authors response: Thanks for these comments and we have included this aspects in the revised version.
Comments on the Quality of English Language
Overall Comments on the Quality of English Language:
The manuscript is generally well-written but requires moderate editing to improve readability and clarity. The main areas needing attention are sentence structure, repetition, word choice, transitions between ideas, and minor grammatical issues. Enhancing these areas will significantly improve the flow of the paper and ensure a more polished and professional tone. Below are examples illustrating the areas that require improvement:
- Many sentences are long and complex, making them harder to follow. Simplifying and shortening these sentences would improve clarity.
Example: "The mortality rates associated with cardiovascular disease and diabetes exhibit disparities by region, with Central Africa ranking fourth globally in terms of mortality rate, this study aimed to determine the death rate attributable to CVD and diabetes in two cities in the DRC."
Suggested Revision: "Cardiovascular disease and diabetes mortality rates vary significantly by region, with Central Africa ranking fourth globally. This study aimed to determine the death rate attributable to CVD and diabetes in two cities in the DRC."
- Certain ideas are repeated unnecessarily, which affects the conciseness of the manuscript. Removing redundant information will improve the flow. Example: "The study found that CVD caused 20.4% of deaths in the two cities. After adjusting for age and city, CVD caused 20.4% of deaths in both Kinshasa and Matadi." Suggestion: "CVD accounted for 20.4% of deaths in the two cities. This proportion remained the same after adjusting for age and city."
- Some terms and phrases could be more precise and scientific. Using more concise and appropriate terminology will improve the manuscript's professionalism. Example: "This data was then cleaned and processed by using the DORIS tool, which is a tool designed for automatic cause-of-death identification." Revision: "The data was cleaned and processed using the DORIS tool, designed for automated cause-of-death identification."
- Flow: The transitions between ideas, particularly in the results and discussion sections, could be smoother. Improved flow will help readers better follow the narrative. Original: "The results indicated a higher mortality rate for women. Cardiovascular diseases were a significant cause of death in the sample. Diabetes was also a major contributor." Suggested Revision: "The results showed that women had a higher mortality rate than men, with cardiovascular diseases being a significant cause of death. Diabetes was also identified as a major contributor."
Authors response: Thanks for these comments, before submitting the manuscript. We have submitted the first version to MDPI English editing service, and We are open to request further edits from the same services
Reviewer 5 Report
Comments and Suggestions for Authors IJERPH (ISSN 1660-4601) Manuscript ID: ijerph-3234274 Cardiovascular Disease and Diabetes among the Main Underlying Causes of Death in Twenty Healthcare Facilities across Two Cities in the Democratic Republic of Congo
Dear Authors,
Both CVD and diabetes are significant causes of mortality in vast majority of locations in the world. Discussion section was well written. At the same time, there was lack of coherence. Refine the whole discussion part.
I have concerns about other areas of the manuscript as well. I am summarizing them below:
Comments:
1. The presentation of these sentences were not that appropriate regarding the quality of the sentences and word choices.
Kindly, rephrase this "These results should also challenge clinicians regarding how they conduct anamnesis, paying particular attention to NCDs both in their management and in raising patients’ awareness of how to behave when affected by these diseases".
2. How these two cities Kinshasa and Matadi, represent broader trends in Central Africa?
3. Add (your) those responses (2) in the introduction section.
4. What about the mortality rates of other regions in Central Africa? Was it the same as these cities? or not?
5. What were the treatments followed by those in the CVD, KD, and diabetes categories (insulin, anti-diabetic meds, etc.)
6. Was there any difference between men and women in those categories in regarding treatments?
7. Mention the full form of abbreviations like DORIS, DRC, HTN, etc.
8. Correct the formatting errors.
9. Mention recent the global prevalence of CVD and diabetes (adult population, men and women) in the introduction section.
10. Add the negative impacts of those diseases in people's quality of life in introduction.
11. This sentence was incomplete. Rewrite that. ""Two experts from the African..............verbal autopsy methods" was incomplete"
12. Given that the findings are disproportionately affected by CVD and diabetes related mortality, what specific interventions could be targeted at women?
13. Fig 1: Both the colour representation of "female" and axis titles were blue, modify any one of them.
14. I would suggest the same change for all figures.
15. Did you mean cities? or projects, here. "Subsequently, four..................................... in two pilot cities"
16. How DORIS tool helped in assessing data quality. What specific parameters were automated?
17. Add a brief note about DORIS role and assistance in data quality assessment.
18. Mention the manual assessments done by three assessors.
19. Did you find hormonal, behavioral, healthcare access, or medication differences between men and women in this study?
20. Did you consider those differences in two cities?
21. How does this study's conclusion about the need for early detection of these diseases align with current Central African healthcare policies regarding NCD prevention?
Comments on the Quality of English Language
Language correction should be done.
Author Response
Both CVD and diabetes are significant causes of mortality in vast majority of locations in the world. The discussion section was well written. At the same time, there was lack of coherence. Refine the whole discussion part.
I have concerns about other areas of the manuscript as well. I am summarizing them below:
Comments:
- The presentation of these sentences were not that appropriate regarding the quality of the sentences and word choices.
Kindly, rephrase this "These results should also challenge clinicians regarding how they conduct anamnesis, paying particular attention to NCDs both in their management and in raising patients’ awareness of how to behave when affected by these diseases".
Authors response: Thanks for these comments, we have edited the manuscript accordingly. This statement was deleted
- How do these two cities Kinshasa and Matadi, represent broader trends in Central Africa?
Authors response: This is a pilot study that does not even represent the complete picture of these two cities (Kinshasa and Matadi), nor of the DRC, and even less so of Central Africa. However, Matadi and Kinshasa share borders with Congo Brazzaville and Angola.
- Add (your) those responses (2) in the introduction section.
Authors response: We did accordingly
- What about the mortality rates of other regions in Central Africa? Was it the same as these cities? or not?
Authors response: We did not get such study, but we found for Tanzania.
- What were the treatments followed by those in the CVD, KD, and diabetes categories (insulin, anti-diabetic meds, etc.)
Authors response: Excellent observation. Nonetheless, within the scope of our study, we are not specifically focused on patient management. However, it is a significant aspect that warrants exploration in future research.
- Was there any difference between men and women in those categories in regarding treatments?
Authors response: The study focused on the causes of death and did not collect data related to treatment, which could be considered one of its limitations.
- Mention the full form of abbreviations like DORIS, DRC, HTN, etc.
Authors response: Thank you for your comment. We have edited the manuscript accordingly
- Correct the formatting errors.:
Authors response: Thank you for your comment. We have edited the manuscript accordingly. The MDPI editing service will also help us for this task. Before submitting the manuscript. We have submitted the first version to MDPI English editing service and We are open to request further edits from the same services
- Mention recent the global prevalence of CVD and diabetes (adult population, men and women) in the introduction section.
Authors response: Thank you for your comment and we have cited the most recent study: “Standing up to infectious disease. Nat Microbiol. 2019 Jan;4(1):1. doi: 10.1038/s41564-018-0331-3. PMID: 30546101; PMCID: PMC7097104”.
- Add the negative impacts of those diseases in people's quality of life in introduction.
Authors response: Thank you for your comment. It has now been taken into account in the introduction section.
- This sentence was incomplete. Rewrite that. ""Two experts from the African..............verbal autopsy methods" was incomplete"
Authors response: This was done
- Given that the findings are disproportionately affected by CVD and diabetes related mortality, what specific interventions could be targeted at women?
Thank you for your comment.
It has now been taken into account in the discussion section.
- Fig 1: Both the colour representation of "female" and axis titles were blue, modify any one of them.
Authors response: We have improved the quality of the figures based on this comment
I would suggest the same change for all figures.
Authors response: We have improved the quality of the figures based on this comment
- Did you mean cities? or projects, here. "Subsequently, four..................................... in two pilot cities"
Thank you for your comment,
These are indeed two pilot cities. They were designated as pilots because the idea was that surveillance of causes of death in the DRC should be extended to other cities up to the national level.
- How DORIS tool helped in assessing data quality. What specific parameters were automated?
Thank you for your comment
The DORIS application has enabled automatic detection of the underlying causes of death. Each subject was assigned three causes of death (immediate, intermediate and underlying). Page 3, lines 32 to 46 of the methodology section explain this better.
- Add a brief note about DORIS role and assistance in data quality assessment.
Thanks for your comments.
Another reference for explain DORIS functions has been added. Methodology section.
- Mention the manual assessments done by three assessors.
Thank you for your comment.
It has now been taken into account in the methodology section
- Did you find hormonal, behavioral, healthcare access, or medication differences between men and women in this study?
Thank you for your comment.
These factors were not particularly addressed in our study. However, we discussed them in the discussion section.
- Did you consider those differences in two cities?
We did not assess hormonal, behavioral, healthcare access, or medication differences between the two cities21. How does this study's conclusion about the need for early detection of these diseases align with current Central African healthcare policies regarding NCD prevention?
Authors response: Thanks for these comments, before submitting the manuscript. We have submitted the first version to MDPI English editing service, and We are open to request further edits from the same services
Reviewer 6 Report
Comments and Suggestions for Authors
Dear Authors
Thanks for splendid research and work, for making the paper more rational presentation please modify some points:
1- In the introduction, the importance of the study is not enough being described and not comparing in the geographical areas sufficiently!
2- some tables and graphs are overlapped , for example tables 2 and 3 by figures 1 , and ,,,,,,
3-The number of references in discussion and introduction parts are not enough, while some valuable up to date recognized ones are available in science direct and other distinguished published papers in recent years.
4- The presenting form of figures can be improved { choose the right form of figures and designing them by more clarity and consistency} and interparty them can be more improved in the manuscript.
Best wishes
Author Response
Comments and Suggestions for Authors
Dear Authors
Thanks for splendid research and work, for making the paper more rational presentation please modify some points:
1- In the introduction, the importance of the study is not enough being described and not comparing in the geographical areas sufficiently!
Authors response: Thanks for the comments, we have edited the introduction and discussion section accordingly. Concerning the details in the regional mortality comparison, we believe that the summary offered is sufficiently concise and precise, especially in the discussion section.
2- some tables and graphs are overlapped , for example tables 2 and 3 by figures 1 , and ,,,,,,
Authors response: Thanks for the comments. We did not have a representative sample of the whole country, that is why we are comparing results by city. Causes of death are associated by the sex and we are comparing by sex that why this looks like some overlapping. Indeed, Table 2 is presenting the distribution of sample by city and Table 3 is presenting the distribution sample by sex. Figure 1 is showing the trend of proportional mortality from communicable diseases (CD) by age and sex. Figure 2 presents the trend of proportional mortality from kidney disease (KD) by age and city. Figure 3 is presenting the trend of proportional mortality from cardiovascular disease by age and sex and Figure 4 is presenting the trend of proportional mortality from diabetes by age and sex.
3-The number of references in discussion and introduction parts are not enough, while some valuable up to date recognized ones are available in science direct and other distinguished published papers in recent years.
Authors response: Thanks for the comment, In the revised version, we have added other references
4- The presenting form of figures can be improved { choose the right form of figures and designing them by more clarity and consistency} and interparty them can be more improved in the manuscript.
Authors response: Thanks for the comment: Figures are improved
Round 2
Reviewer 3 Report
Comments and Suggestions for Authors
I have confirmed that the author has made appropriate corrections to the manuscript in response to my previous comments.
Reviewer 5 Report
Comments and Suggestions for Authors
Dear Authors,
I could see that you have improved your manuscript. All the best.
Comments on the Quality of English LanguageIt's better.
Minor editing will enhance the quality of the manuscript.